# Evolution of Heart Rate Variability and Heart Rate Turbulence in Patients with Depressive Illness Treated with Selective Serotonin Reuptake Inhibitors

**DOI:** 10.3390/medicina56110590

**Published:** 2020-11-05

**Authors:** Catalina Giurgi-Oncu, Cristina Tudoran, Virgil Radu Enatescu, Mariana Tudoran, Gheorghe Nicusor Pop, Cristina Bredicean

**Affiliations:** 1Discipline of Psychiatry, Department of Neuroscience, University of Medicine and Pharmacy “Victor Babes” Timisoara, Eftimie Murgu Place Nr. 2, 300041 Timis, Romania; catalina.giurgi@umft.ro (C.G.-O.); enatescu.virgil@umft.ro (V.R.E.); bredicean.ana@umft.ro (C.B.); 2Discipline of Cardiology, Department VII, Internal Medicine II, University of Medicine and Pharmacy “Victor Babes” Timisoara, Eftimie Murgu Place Nr. 2, 300041 Timis, Romania; tudoran.mariana@umft.ro (M.T.); pop.nicusor@umft.ro (G.N.P.)

**Keywords:** major depressive disorder, selective serotonin reuptake inhibitors, heart rate variability, heart rate turbulence, arrhythmias

## Abstract

*Background and objectives*: Major depressive disorder (MDD) and cardiovascular diseases (CVDs) represent serious and prevalent medical conditions. Autonomic nervous system (ANS) dysfunctions, expressed by parameters of heart rate variability (HRV) and heart rate turbulence (HRT), have been repeatedly associated with depression. The aim of our study was to identify potential HRV and HRT alterations in patients with MDD, before and after selective serotonin reuptake inhibitor (SSRI) therapy, and to observe any correlations between these parameters and the severity of depressive symptoms. Another aim was to evidence if there was a normalization or improvement of HRV and HRT parameters that paralleled the reduction of the intensity of depressive symptoms. *Materials and Methods*: We studied heart rate variability (HRV) and heart rate turbulence (HRT) in a sample of 78 patients, aged under 51 years, who were referred to our outpatient clinic between June 2017 and December 2019, for complaints in the context of a new onset major depressive disorder (MDD), before and after therapy with SSRIs. *Results*: By using 24 h Holter ECG monitoring, we evidenced alterations of HRV and HRT parameters, significantly correlated with the severity of depressive symptoms (*p* < 0.001), as assessed by the Montgomery–Åsberg Depression Rating Scale (MADRS). Our results indicated that these parameters improved following one and six months of SSRI therapy, when a partial or total remission of depressive symptoms was achieved (*p* < 0.001). Changes in HRV parameters were correlated with the reduction of the severity of depression. *Conclusions*: In our study group, we highlighted, through 24 h ECG Holter monitoring, frequent and clear HRV and HRT abnormalities that were statistically correlated with the severity of depressive symptoms. Furthermore, we were able to document a progressive improvement of these parameters, which corresponded with the improvement of depressive symptoms under SSRI therapy, when compared to the values measured before the commencement of antidepressant pharmacotherapy.

## 1. Introduction

In modern society, a current epidemic is represented by depressive illness, with the World Health Organization (WHO) estimating this to become the leading cause of disability and a major factor in the global burden of disease by 2020 [1]. Irrespective of geographical location, major depressive disorder (MDD) and cardiovascular diseases (CVDs) represent two of the most severe public health challenges in contemporary society [2]. It has been well established that there are significant correlations between these afflictions, with more unfavorable prognoses for people diagnosed with comorbid depression and CVD [3,4]. On its own, MDD is associated with a high risk of developing atherosclerosis and heart failure, while in patients with previously diagnosed coronary artery disease (CAD), arrhythmias, or heart failure, comorbid depressive symptoms represent an unfavorable prognosis factor [5]. Moreover, there are several pathophysiological pathways shared by the two illnesses [6]; of these, an essential one is represented by the dysregulation of the hypothalamic–pituitary–adrenal axis, which gives way to an increased release of cortisol and norepinephrine [7]. Additionally, both depressive illness and cardiovascular disease result from a sympathoadrenal hyperactivity; an amplified response of the sympathetic nervous system (SNS) and the inhibition of the parasympathetic nervous system (PNS) lead to an imbalance of the autonomic nervous system (ANS), and its dysregulation is considered to represent a potential pathophysiological mechanism in the onset of depression [8]. Cumulated with the sympathetic hyperactivity, the parasympathetic withdrawal, translated into a lower relaxation response, may also lower the threshold for myocardial ischemia and ventricular arrhythmias, and potentially predispose an individual to sudden cardiac death [5]. For the purposes of correctly identifying and evaluating the well-being status via alterations of the ANS balance, changes in heart rate variability (HRV) and heart rate turbulence (HRT) represent fairly new, non-invasive, and accurate indicators [7,8,9]. HRV monitoring is a marker that identifies the physiological spontaneous fluctuations in heart rate (HR) and normal R-R intervals and is currently being widely used to characterize the status of the underlying ANS [7,8]. Its analysis is used in both physiological models, as well as in various pathological states, for the assessment of cardiovascular risk. HRT is a method that studies the sinus rhythm cycle length variation, following isolated premature ventricular contractions (PVCs). Certain authors have identified data suggesting that an increased HR, due to sympathetic hyperactivity, represents an accurate biomarker and a negative prognostic factor for CVD [10,11]. The sympathovagal imbalance may induce arrhythmias and increase the risk of sudden cardiac death [12,13]. Moreover, a reduced HRV and an altered HRT have been associated with the severity of depressive symptoms, while changes in their respective parameters appear to be related to the evolution of depressive illness [8,14,15].

With regard to depression, current international guidelines highlight selective serotonin reuptake inhibitor (SSRI) antidepressant therapy as the first-line psychotropic treatment for moderate/severe depression, with significant benefit for the evolution of patients with MDD [16]. In this drug class, to date, sertraline is still considered to be the gold standard treatment option, owing to its safety in the treatment of patients with comorbid CVD [16,17,18,19]. It has been shown that SSRIs influence cardiovascular functioning through several mechanisms; for example, they inhibit serotonin-mediated and collagen-mediated platelet aggregation, as well as reducing inflammatory mediator levels. SSRIs improve cardiac function in CAD and heart failure, without adversely affecting electrocardiographic parameters. To date, results of studies concerning the association between HRV and HRT parameters and the response to antidepressant treatment [8,13,16,18,19] are varying, most studies suggesting beneficial effects on the SNS and PNS dysregulation.

Starting from the hypothesis that, in depressed individuals, the SNS and PNS activation or inhibition, as well as their magnitude, could be reflected by fluctuations of HRV and HRT parameters and that antidepressant therapy could influence these processes, the aim of our study was to identify potential HRV and HRT alterations in this patient group, before and after SSRI therapy, and to observe any potential correlations between these parameters and the severity of depressive symptoms. A second aim was to document if there is a normalization or improvement of HRV and HRT parameters that parallels the reduction of the intensity of depressive symptoms following one and six months of SSRI pharmacotherapy.

## 2. Materials and Methods

### 2.1. Study Group

We selected a group of 97 patients, aged under 51 years, from the cohort of patients that had been referred to the outpatient psychiatric service of the Clinical County Emergency Hospital “Pius Brinzeu” Timisoara, Romania, for depressive symptoms, from June 2017 to December 2019. The inclusion criteria required that all patients be diagnosed with a first episode of major depression, as per the Diagnostic and Statistical Manual of Mental Disorders, 5th Edition (DSM-5) [20], and not be prescribed antidepressant treatment prior to or at the moment of inclusion in the study. Antidepressant therapy with sertraline 25–50 mg/day was initiated, past the point of the initial cardiology evaluation. Subsequently, most patients were maintained on a dose of sertraline between 50 and 200 mg/day. We excluded all subjects that had been previously diagnosed with CVD, diabetes mellitus, dyslipidemia, or with significant risk factors for atherosclerosis (including smoking). Considering the intensity of depressive symptoms, we divided all patients into the following subgroups: mild, moderate, or severe depression. Of all patients, 19 did not tolerate or respond to this particular SSRI therapy, and were thus excluded from our study, moving on to receive other psychopharmacological antidepressant treatment strategies. It is worth mentioning that all of the patients who were initially recruited for our study received psychoeducation at the beginning of the intervention. As it is not offered by the Romanian public health system, formal individual talk therapy was not available, as part of the treatment strategy. The remaining 78 patients (29 men and 49 women) were evaluated after a month of therapy, when a significant improvement of depressive symptoms was achieved and, once more, after six months of therapy, when there was an evident remission of depressive symptoms.

### 2.2. Mental Health Evaluation

Mental health complaints were assessed by specialist psychiatrists, who evaluated the intensity of current depressive symptoms in all patients who had been referred by their general practitioner. The diagnosis of major depression was confirmed following a clinical interview that assessed for the presence of recent depressed mood or anhedonia (loss of interest or pleasure) for a minimum of 2 weeks prior to the referral, accompanied by significant functional impairment and additional somatic or cognitive symptoms (fatigue or loss of energy, sleep or appetite disturbance, psychomotor agitation or inhibition, concentration difficulties, feelings of worthlessness or hopelessness, suicidal ideation), as per the DSM-5 criteria [20].

For the purposes of a more objective evaluation of mental health complaints, the Montgomery–Åsberg Depression Rating Scale (MADRS) was also used. This is a 10-item rating scale that aims to evaluate the severity of depression based on the total score, with higher results indicating a greater severity of depression [21]. Accordingly, we classified the patients into the following three subgroups, depending on the severity of their current depressive symptoms: mild, moderate, and severe. The MADRS has high inter-rater reliability and correlates significantly with scores of other standard scales for depression, such as the Hamilton Depression Scale (HAM-D) [22]. For each patient, we also employed a comprehensive sociodemographic questionnaire. A psychoeducation intervention was offered to all participants upon inclusion in the study.

### 2.3. Cardiology Evaluation

After a rigorous cardiovascular examination that included individual medical history, physical examination, and 12-lead ECG, all patients were evaluated by echocardiography. For the purposes of avoiding inter-observer differences, the same experienced operator performed all assessments, according to current guidelines’ recommendations. We determined the dimensions and function of cardiac cavities and the ejection fraction (EF) by using the Simpson method. All patients were fitted with a 24 h Holter Labtech Cardiospy device monitor (Labtech Ltd, Debrecen, Hungary). The cardiology team analyzed the occurrence of ventricular arrhythmias and classified them according to their severity. The Nevrokard Long-Term aHLV (L-aHRV V.5.0.0., Nevrokard Kiauta, Izola, Slovenia) program was used to analyze the obtained data and provide an accurate HRV analysis, which automatically excluded all premature atrial contractions (PACs) and premature ventricular contractions (PVCs) from the 24 h ECG recordings. All ECG recordings were subsequently inspected to identify potential artefacts that could have been generated by electrode displacement or movement and were compared with individual patients’ records of events, activity, or sleep. The artefacts that were not detected by the filter were deleted manually. Regarding HRV, which represents the physiological phenomenon of variation in the time interval between consecutive heartbeats, we assessed the standard deviation of all normal to normal (NN) intervals (SDNN) and the root mean square of successive differences between normal heartbeats (RMSSD) in the time domain, and in the frequency domain, high frequency (HF). Patients with SDNN values below 50 ms were classified as unhealthy, those with values in the range between 50 and 100 ms were deemed as showing a compromised health status, while those who obtained values above 100 ms were considered to be in a perfect state of health [23,24]. In order to determine HRT, we only included patients with at least 6 PVCs. For HRT, which analyzes the sinus rhythm cycle length variation following isolated premature ventricular contractions (PVCs), we determined the turbulence onset (TO), i.e., early sinus acceleration after a PVC, and the turbulence slope (TS), i.e., late sinus deceleration following a PVC. In most studies, TO < 0% and TS > 2.5 ms/R-R interval are considered normal, but according to risk stratification studies, HRT can be divided into the following 3 categories: category 0, TO and TS are normal; category 1, TO or TS is abnormal; and category 2, TO and TS are abnormal [23,24].

All investigations were repeated after one month and six months of SSRI therapy.

### 2.4. Statistical Methods

Data analysis was performed using the Statistical Package for the Social Sciences v.25 (SPSS, Chicago, IL, USA). Continuous variables were presented as mean and standard deviation (SD) or median and interquartile range (IQR), and categorical variables were presented as frequency and percentages. Results of the normality test (Shapiro–Wilk) showed a non-Gaussian distribution, suggesting that we continue the analysis by using nonparametric tests. A chi-squared trend test was employed to evaluate the significance of the differences in the proportions of premature beats and category over time. For the evaluation of a potential connection between the severity of depression and HRT, we used Spearman’s correlation test. Friedman’s Two-Way Analysis of Variance by Ranks for Related Samples was employed to compare the evolution over time (initially, at one month, at six months) of continuous variables (MADRS score, SDNN, RMSSD, HF, TO, and TS values), followed by the Wilcoxon test for pairwise comparison with *p*-value, adjusted using the Bonferroni correction. We considered a *p*-value of under 0.05 to indicate a statistically significant difference.

The Local Scientific Research Ethics Committee of our hospital approved the design and methodology of the study (No. 15/31.05.2017). All of the patients whom we included in the evaluation signed individual informed consent forms prior to data collection.

## 3. Results

At the beginning of the study, of the 78 patients (29 men and 49 women) with depressive illness who were yet to be prescribed psychotropic medication, 23 patients reported severe symptoms (scores 35 to 60), 31 patients were noted as suffering from a moderate form (scores 20 to 34), while 24 patients had mild complaints (scores 7 to 19). The age category of our selected patient population was relatively young, with ages between 31 and 51 years, and a median age of 41.5 (40 to 46). The remission of depressive symptoms was obtained within 1-3 months, after an average of two months of SSRI therapy. The mental health status of patients was evaluated by means of a specialist clinical interview aided by scores on the MADRS, while their physical health status was analyzed by echocardiography and 24 h Holter monitoring, at one and six months after the initial referral. At the final evaluation time point, all patients who responded to sertraline were symptom free for at least two months (however, they still required SSRI maintenance therapy), as presented in Table 1.

Results from the 24 h ECG Holter monitoring indicated that all patients had an increased heart rate, more frequent premature ventricular contractions (PVCs), including unsustained ventricular tachycardia (VT) in two cases, in association with reduced values of SDNN and alterations of HRT (Table 1). At the beginning of our study, and before starting antidepressant therapy, all patients showed evidence of an ANS imbalance, as expressed by lower or even pathological values of SDNN (under 50 ms), RMSSD and HF, as well as a pathological HRT. At this time point, the statistical analysis evidenced strong significant negative correlations between scores of the MADRS scale and the values of SDNN and RMSSD (r = −0.88, 95% CI (−0.942; −0.787), respectively, r = −0.709, 95% CI (−0.846; −0.520), *p* < 0.001) and moderate ones with the HF (r = −0.505, 95% CI (−0.673; −0.288), *p* < 0.001). After the first month of therapy, and when in complete remission of depressive symptoms, these correlations were moderate for SDNN (r = −0.692, 95% CI (−0.813; −0.534), respectively, r = −0.729, 95% CI (−0.844; −0.563)) and RMSSD (r = −0.637, 95% CI (−0.770; −0.461), respectively, r = −0.424, 95% CI (−0.626), but still statistically significant *p* < 0.001.

Considering the evolution of SDNN during SSRI therapy, we noticed the largest increase, at 54.82%, after the first month of treatment, in parallel with the reduction of MADRS scores (*p* < 0.001), see Figure 1. This tendency continued at six months, with a lesser (4.16%), but nonetheless statistically significant, level (*p* < 0.001), see Figure 1. A similar evolution was observed for RMSSD, while HF had a constant, if smaller, hence statistically significant (*p* < 0.001) increase.

With reference to HRT, we initially evidenced a significant correlation between TO and the MADRS score (r = 0.771, 95% CI (0.653; 0.855), *p* < 0.001), and moderate associations after the first month of therapy (r = 0.600, 95% CI (0.395; 0.747), *p* < 0.001) and at six months (r = 0.577, 95% CI (0.387; 0.728), *p* < 0.001). Concerning TS, we initially evidenced a weak, but significant, correlation with the MADRS score (r = −0.453, 95% CI (−0.633; −0.260), *p* < 0.001), and, similarly, at one month (r = −0.24, 95% CI (−0.454; −0.017), *p* = 0.03) and at six months (r = −0.375, 95% CI (−0.555; −0.162), *p* = 0.001). Of the entire group, only 10 subjects initially had a normal TO and TS, which placed them in category 0, while 23 patients had alterations of only one parameter, thus placing them in category 1; the remaining 45 patients had pathological levels of both TO and TS, thus placing them in category 2. We repeated these evaluations after six months of SSRI therapy, when all patients were symptom free for at least two months, and noticed that 36 subjects were now in category 0, 21 patients belonged in category 1, while the remaining 21 had both TO and TS still ranging in the pathological domain, thus placing them in category 2. At this stage, we demonstrated a statistically significant improvement of all studied parameters (*p* < 0.001), as noted in Table 1 and Figure 1.

Upon analysis of the HRT evolution, we observed that the TO had a maximum decrease after the first month of SSRI therapy, followed by a soft slope at six months, while TS showed a slow, progressive growth. Accordingly, these differences were statistically significant (*p* < 0.001), as seen in Figure 1.

## 4. Discussion

The effect of major depression on the ANS, potentially leading to HRV and HRT alterations, has been a highly debated topic in recent studies [4,7,13]. Several research findings evidenced a reduced HRV, as expressed by lower values of SDNN, for patients suffering from comorbid depression, when compared to healthy controls [7]. Moreover, a reduced HRV has been associated with the severity of depression, and the derived parameters have been used to delineate its severity or even changes in symptom severity [19]. Several studies including patients with CVD [24] have documented increased heart rate values and reduced HRV in subjects with comorbid depression, as compared to those without mental health complaints. However, the association between depression and a reduced HRV (or other ANS dysregulation parameters) is not entirely consistent across all studies. Part of these effects may be determined by antidepressant medications [5].

Taking into account that HRV is expected to decrease as we age, and in order to limit the impact of this factor on our results, we selected younger subjects, under the age of 51 years, newly diagnosed with a depressive episode, thus without prior exposure to antidepressant therapy. There are debates in the medical literature over the consequences of depression on HRV, in relation to the patient’s age, with potential variations across different age cohorts. Age-related decline may lower HRV to levels associated with an increased risk of mortality; therefore, an effort to accurately distinguish a low HRV caused by comorbid depression from that due to normal aging is challenging. It has been suggested that depression in older persons may additionally intensify age-related declines in HRV, thus exacerbating the risk of cardiovascular morbidity [16]. In our research, a reduced HRV was constantly detected among patients with clinical depression, and a strong negative correlation between the severity of depressive illness and SDNN was highlighted. Similar results were reported in several studies, evidencing the potential role of HRV as a biomarker for the severity of depression. Carnevalli et al. [8] investigated the relationship between HRV parameters and the severity of symptoms, aiming to test the hypothesis that changes in HRV parameters correlate with changes in the severity of depression, and evidencing reciprocal correlations for depression scales and HRV.

Existing research data on the association between HRV and the response to antidepressant treatment offer varying results [16,18,19]. Research studies that include a longitudinal examination of HRV alongside an accurate monitoring of the severity of depressive symptoms are scarce. Nevertheless, some findings suggest changes in HRV that appear to normalize in line with a notable improvement in the severity of clinical depression during antidepressant treatment [12]. Hartman et al. evidenced a normalization of HRV parameters, equivalent to an improvement of depressive symptoms, after two weeks of antidepressant treatment [7]. In their study, Carnevalli et al. [8] also documented a normalization or significant improvement of HRV parameters after two weeks of antidepressant therapy. The authors suggested that this could be due to a dysregulation of the two ANS branches, which may be a shared pathophysiological mechanism, alongside changes in the arousal regulation during therapy that may relate to changes in the severity of depression. In our study, we also noticed statistically significant improvements of HRV and HRT parameters in all of our patients, following the remission of depressive symptoms under SSRI pharmacotherapy, starting with the first month of therapy and continuing at six months (*p* < 0.001).

ANS imbalance is associated with major depression, and alterations of HRV and HRT were frequently evidenced for this patient category in various medical studies [25,26,27]. It seems that the amplitude of HRV reduction matches the severity of depressive symptoms. Data regarding the reversibility of HRV and HRT alterations under therapy with antidepressants are not entirely conclusive, with some studies evidencing further reductions of HRV during treatment. Nevertheless, in the case of SSRI therapy, notable improvements were evidenced in multiple studies [25,26,27,28]. However, we have highlighted the importance of early detection of mental health difficulties, when aiming to prevent and even treat CVD. It is expected that better overall clinical results, an increased individual quality of life, and decreased health expenditures will result when both somatic and mental health clinicians will pay attention to the early and correct identification of these comorbidities, by working collaboratively and holistically. We suggest a need for the development of wider concerted programs, in order to accurately detect CVD comorbidities in patients with clinical depression, in an effort to develop timely rounded intervention plans for this patient group.

### Study Limitations

First of all, we did not have a control group available for the comparison of results, which meant that we solely relied on normal values’ data published in specialized literature. Another limitation was that, in order to respect our selection criteria, we did not use a randomized group of patients. A third limitation was the size of our study group, which was too small to garner clinically relevant conclusions. Thus, these limitations may have impacted the significance of our results.

## 5. Conclusions

Alterations of the ANS are a common finding among patients suffering from depressive illness. In our study group, we evidenced, by using 24 h ECG Holter monitoring, frequent and clear HRV and HRT abnormalities that were statistically correlated with the severity of depressive symptoms, as assessed by means of a clinical interview and the MADRS scale. Furthermore, we documented a progressive improvement of HRV and HRT parameters in our patients with comorbid depression, corresponding with the improvement of depressive symptoms under SSRI therapy, compared to the individual values, as assessed before the commencement of antidepressant pharmacotherapy.

## Figures and Tables

**Figure 1 medicina-56-00590-f001:**
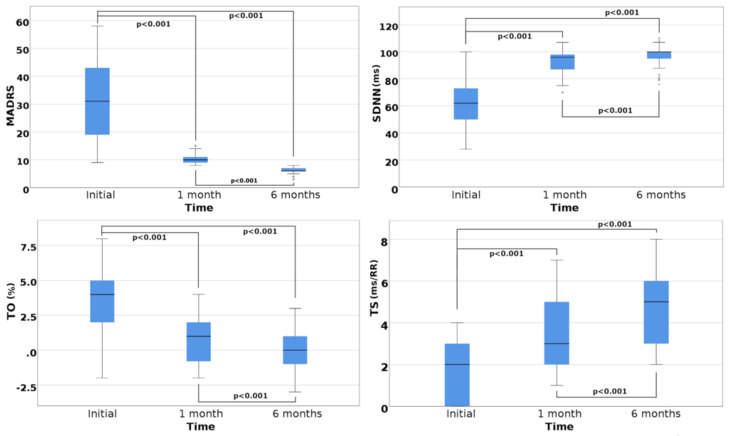
Evolution of Montgomery–Åsberg Depression Rating Scale (MADRS) scores, heart rate variability (HRV) and heart rate turbulence (HRT) parameters. Legend: SDNN, standard deviation of all normal to normal (NN) intervals; TO, turbulence onset; TS, turbulence slope; Wilson test for pairwise comparison with *p*-value adjusted using the Bonferroni correction.

**Table 1 medicina-56-00590-t001:** Results of clinical data, 2D echocardiography, and 24 h Holter monitoring.

MADRS Scale, LVEF, and Results of 24 h ECG Holter Monitoring	T0	T1	T6	*p*-Value
MADRS scores	31 (19–43)	10 (9–11)	6 (6–7)	<0.001
Echocardiography: LVEF (Simpson) ^b^	61 (59–62)	65 (63–66)	66 (64–68)	<0.001
24 h ECG Holter monitoring:				
Mean heart rate ^b^	79 (78–81)	64 (63–66)	63 (62–64)	<0.001
Premature supraventricular beats ^a^	49	33	21	<0.001
PVC: isolated ^a^	43	29	24	0.193
systematized ^a^	28	13	9	<0.001
unsustained TV ^a^	2	-	-	-
HRV: SDNN ^b^	62 (49–74)	96 (87–98)	100 (95–100)	<0.001
RMSSD ^b^	27.7 (27.2–28.9)	29.6 (29–30.1)	30.7 (30.2–31)	<0.001
HF ^b^	240 (227–263)	279 (267–290)	311 (298–321)	<0.001
HRT: TO, %	4 (2–5)	1 (−0.8–2)	0 (−1–1)	<0.001
TS, ms/RR	2 (0–3)	3 (2–5)	5 (3–6)	<0.001
Category 0 ^a^	10	21	36	<0.001
Category 1 ^a^	23	33	21	0.071
Category 2 ^a^	45	24	21	<0.001

T0, Before SSRI treatment; T1, After 1 month of therapy; T6, After remission of depression; SSRI, selective serotonin reuptake inhibitors; MADRS, Montgomery–Åsberg Depression Rating Scale; LVEF, left ventricular ejection fraction; ECG, electrocardiogram; HRV, heart rate variability; SDNN, standard deviation of all normal to normal (NN) intervals; RMSSD, Root Mean Square of Successive Differences between normal heartbeats; HF, high frequency; HRT, heart rate turbulence; TO, turbulence onset; TS, turbulence slope. ^a^ number (frequency), chi-squared trend test; ^b^ median (Q1-Q3), Friedman’s Two-Way Analysis of Variance by Ranks for Related Samples.

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
