# Peer review of "Evolution of Heart Rate Variability and Heart Rate Turbulence in Patients with Depressive Illness Treated with Selective Serotonin Reuptake Inhibitors"

_medicina, 2020, doi:10.3390/medicina56110590_

Round 1
Reviewer 1 Report
This paper addresses an important topic relating to changes in Heart Rate Variability associated with Major Depressive Disorder and its treatment with an antidepressant medication.
The inclusion of Heart Rate Turbulence as a parameter is of interest and the 24 hour monitoring of ECG activity, valuable. However there are significant limitations in the design, some of which the authors recognise. I suggest there are a number of problems which need to be addressed:
1) Why was the range of doses of sertraline employed so low? A maximum of 100mg per day is unusual in a sample which includes "severely depressed" patients.
2) Why were the 19 patients who did not respond to sertraline (noting the relatively low maximum dose), not studied as a "control" group for non-specific factors as participants? There is no mention of possible response to factors such as the participation in a research project, with the emotional support that entails. Was psychotherapy of any kind employed during the six month study period? Were all other medications excluded?
3) The diagnostic criteria of MDD employed in this study were not fully explained. Did the assessing psychiatrists use ICD or DSM criteria, for example?
4) The MADRS is a well established rating scale for depression but appears to have been the only measure of psychological symptoms/signs employed. While the authors described the MADRS in considerable detail (more than I think is required), its limitations are not mentioned. Rating scales are invaluable but are usefully supplemented by questionnaires or visual analogue scales to offer a patients' subjective experience and further information. Correlations of HRV and HRT with individual item scores would have been of interest, but the significance limited by the sample number.
5) The reporting of the SDNN parameter for HRV alone is very surprising. The SDNN is a valuable measure but reflects a combination of sympathetic and parasympathetic activity. The results of this study would be greatly enhanced through reporting of another time measure (eg RMSSD) and a frequency measure (eg HF).
6) The statistical analysis appears confusing (although I am not expert in this field). The authors describe a non-Gaussian distribution of their data but report continuous variables. Were the distributions of all "apparently" continuous variables non-Gaussian? The suggestion is that a variety of non-parametric tests were employed but the presentation (eg MADRS scores) is confusing.
7) The lack of reporting of standard deviations numerically (although evident graphically) and effect sizes is unhelpful.
8) It was interesting to see that the standard deviations of SDNN fell dramatically with treatment and remission. A comment upon this change in variability would be of interest.
This study contains some valuable data but needs to be reported with greater detail in the areas outlined above and perhaps less detail regarding the virtues of sertraline. I agree it is well regarded as an antidepressant and does not impair cardiac function but this could be summarised much more concisely. Similarly the description of the MADRS could be shortened considerably without losing recognition that this is a well validated measure of depressive illness. I suggest that the scoring system does not need detailed explanation in this context. Addressing these two components of the paper would allow addition of the matters I have raised above, without the paper becoming very much longer.
Author Response
Dear reviewer,
Thank you for all your efforts and time spent to review our manuscript. According to your comments, we revised it and made the following changes: Thank you for these suggestions. We have taken all of it into account and made the required changes.
- Why was the range of doses of sertraline employed so low? A maximum of 100mg per day is unusual in a sample which includes "severely depressed" patients.
Thank you for your observation. This was an oversight in the writing of the manuscript, and it has now been amended to include the correct range of doses.
- Why were the 19 patients who did not respond to sertraline (noting the relatively low maximum dose), not studied as a "control" group for non-specific factors as participants? There is no mention of possible response to factors such as the participation in a research project, with the emotional support that entails. Was psychotherapy of any kind employed during the six-month study period? Were all other medications excluded?
Thank you for your observation. For the purposes of a homogenous study group, we only took into account those patients who responded to a specific SSRI, namely sertraline. We excluded all other treatment strategies. We changed the treatment of the 19 persons who did not tolerate or did not respond to sertraline to a different SSRI or a combination psychopharmacological treatment strategy. We offered psychoeducation to all participants, including the 19 non-responders. Unfortunately, other more specific types of talk therapies are not available in the public health system free of charge.
- The diagnostic criteria of MDD employed in this study were not fully explained. Did the assessing psychiatrists use ICD or DSM criteria, for example?
Thank you for your observation. The assessing psychiatrists used DSM-5 criteria and this has now been described in more detail in the manuscript.
- The MADRS is a well-established rating scale for depression but appears to have been the only measure of psychological symptoms/signs employed. While the authors described the MADRS in considerable detail (more than I think is required), its limitations are not mentioned. Rating scales are invaluable but are usefully supplemented by questionnaires or visual analogue scales to offer a patients' subjective experience and further information. Correlations of HRV and HRT with individual item scores would have been of interest, but the significance limited by the sample number.
Thank you for your observation. We assessed each patient’s mental health by means of a lengthy and comprehensive clinical interview and also by the MADRS, in an effort to acquire objective numerical values that could be further compared throughout the evolution of symptoms under therapy.
- The reporting of the SDNN parameter for HRV alone is very surprising. The SDNN is a valuable measure but reflects a combination of sympathetic and parasympathetic activity. The results of this study would be greatly enhanced through reporting of another time measure (eg RMSSD) and a frequency measure (eg HF).
Thank you for these suggestions. We have taken all of it into account, and we completed our data with the analysis of other parameters of HRV, namely RMSDD and HF.
6) The statistical analysis appears confusing (although I am not expert in this field). The authors describe a non-Gaussian distribution of their data but report continuous variables. Were the distributions of all "apparently" continuous variables non-Gaussian? The suggestion is that a variety of non-parametric tests were employed but the presentation (eg MADRS scores) is confusing.
There are three major types of variables: numerical (discrete/continuous), ordinal and nominal. A numerical variable can follow two patterns of distribution. A normal (Gaussian) distribution is a bell-shaped frequency distribution curve. Most of data values in a normal distribution tend to cluster around the mean. The further a data point is from the mean, the less likely it is to occur. There are many things, such as intelligence, height, and blood pressure, that naturally follow a normal distribution. Normal distributions are symmetric, unimodal, and asymptotic, and the mean, median, and mode are all equal. When one or more of this conditions are violated then we have a non normal (non-Gaussian) distribution (eg. bacteria growth naturally follows an exponential distribution; rare events such as number of accidents; data such as survival times of a product). In conclusion a numerical variable can have a non-Gaussian distribution and, in this case, there are two options: employ non-parametric test (preferred) or transform data, forcing to fit a normal distribution (is used when the test needs a normal distribution and there are no non-parametric test alternative)
7) The lack of reporting of standard deviations numerically (although evident graphically) and effect sizes is unhelpful.
The mean and standard deviation are reported only for normal distribution variables, because the bell-shaped curve is derived from a function and have the propriety that 68.2% of values in the distribution are within 1 SD of the mean, 95.4% of values in the distribution are within 2 SD of the mean, 99.7% of values in the distribution are within 3 SD of the mean, etc. When we deal with numerical variables with non-Gaussian distribution it’s recommended to report median (Q2) and interquartile range (Q1-Q3 ---25% of the scores have a value lower than Q1 and 25% of the scores have a value larger than Q3).
8) It was interesting to see that the standard deviations of SDNN fell dramatically with treatment and remission. A comment upon this change in variability would be of interest.
The standard deviation is defined as the spread of data around its mean. As the mean decreases, there is a possibility that the standard deviation, variance and percentile values remain unchanged. But, in our case, there was a left shift of all the data, and because SDNN could not be lower than zero, the space for data distribution on OX axis is narrowed and for this reason the change of standard deviation happened.
Best regards,
The authors
Reviewer 2 Report
It was a pleasure reading this well written paper and important research. I hope the following comments/suggestion are helpful.
Major Comments:
The idea of ANS balance or imbalance is used throughout. I completely understand what the authors are driving at here, however, many HRV experts disagree with this framing. Though this is a conceptually attractive way to think about the relationship between the PNS and SNS as they act on the heart, it does not really reflect how the branches of the ANS actually work. If the PNS and SNS worked on a principle of balance, as one went up the other would decrease, and vice-versa. In reality they are quite distinct in their action. For example, the human heart is tonically sympathetically engaged, while parasympathetic innervation of the heart acts as a brake. While I appreciate the attractive heuristic of ‘balance’, perhaps as a field would should be using more accurate language that talks about sympathetic/parasympathetic activation/inhibition etc, rather than autonomic balance.
I was expecting to see some hypotheses toward the end of the introduction. Did the authors have any a priori hypotheses? If so, it would be good to mention them.
It seemed odd to cite a single empirical paper (Carnevali et al.) to support the statement, “The results of studies concerning the association between HRV/HRT parameters and the response to antidepressant treatment [8] remain still inconsistent.” There’s a fairly substantial literature on HRV/SSRIs. Perhaps there’s a review paper or meta-analysis on the topic that would make more sense to cite.
It is certainly sound to use SDNN, which reflects overall HRV, but it seems like the manuscript would be strengthened by including a complementary HRV index speaking to parasympathetic activation like pNN50 or RMSSD.
With regard to the post-processing of the raw ECG signal, I wondered if any manual inspection of the raw recordings was done. Automatic filters have come a long way, but they are still highly prone to problems. How, for instance, did the filter used identify and handle segments of noise/artifact arising from movement.
Relatedly, was movement controlled for as a covariate in the models? Also, was sleep time controlled for over the 24 period?
In the statistical methods section I found myself getting lost. It would really help if the dependent and independent variables were clearly stated along with mention of any covariates. e.g., it is stated that, “For the evaluation of the potential connection between MDD and HRT, we used Spearman’s correlation test.” If everyone in the sample had MDD this statement doesn’t make sense. Do the authors mean MDD severity? If so, this should be clearly noted along with the measure being used to capture MDD severity. Later it is stated that, “Related-Samples Friedman’s Two-Way Analysis of Variance by Ranks was employed to compare continuous variables evolution over time…”, but it is not clear what continuous measures are being referred to. Additionally, if there were no covariates, it would be good to mention why important factors like age and potentially sex were not controlled for in the analyses.
In Table 1, what are the number in parentheses? Relatedly, I had a lot of trouble reading Table 1 as the lines were jumbled, which perhaps happened when the document was converted to PDF.
In the results section it is stated that, “Results from the 24 hours ECG Holter monitoring indicated that patients with MDD had an increased heart rate…” Forgive me if I’m misunderstanding something, but didn’t all participants have MDD? The way this sentence is written it sounds like there was a control group.
I may have missed it, but how were the cut points for SDNN ‘health’ determined?
Because I had a little trouble understanding the statistical approach, I had a trouble interpreting some of the results. For instance, some of the r’s reported in the results section are incredibly high (e.g., the correlation between TO and the MADRS; r = .77). Perhaps though I’m misunderstanding the analytic approach and these r’s a plausible.
Minor Comments:
The acronym SSRI is not defined in the body of the manuscript.
It is stated that, “At the final evaluation time point, all patients were symptom-free for at least 2 months, however still required SSRI maintenance therapy, table 1.” Perhaps rephrase this sentence. It’s an odd thing to say given 19 participants were excluded from the study because they were non-responders.
Need to add the units of measurement to the graphs in Figure 1.
Author Response
Dear Reviewer,
Thank you for your time and efforts to review our manuscript. We appreciated your suggestions and tried to apply them as well as possible:
In the introductive part, we explained that, commonly, in major depression, the autonomic nervous system dysregulation refers to the enhancement of sympathetic nervous system (DNS) and inhibition of the parasympathetic nervous system (PNS), thus, the amplitude of these fluctuations is very difficult to be estimated. We also defined the acronym for SSRI.
In the second paragraph, we completed the citations with some other studies over antidepressants effects on HRV/HRT, and in the third paragraph we mentioned the hypothesis which was the origin of our study.
Thank you for the suggestion we complete our data with other parameters of HRV, namely RMSSD and for the frequency domain HF. As response to your concerns, we always inspect manually the ECG recordings for identifying artefacts generated by movement or electrode displacement and compare them with patients’ records on events, activity sleep. The artefacts which were not detected by the filter, were deleted manually. The cut points for SDNN health were determined according to Bauer’s et al results, reference 23, 24.
Regarding statistical methods, this section has been revised. According to your suggestion, we now mentioned the severity of MDD when we referred to the correlation between MDD and HRV. Likewise, we stated which continuous variables were followed over time, namely SDNN, RMSSD, HF, TO and TS. Age and gender were not found as confounding factors in this study, reason why we did not find it necessary to perform any adjustment for age or gender. The small range of age of our patients (between 31 to 51 years old) may be a reason why there was no statistically difference of HRV parameters between age groups.
Sorry for the inconvenience of table 1. We tried to correct its aspect and completed it. All values marked with b are expressed as median and the numbers in parentheses are quartiles. As you suggested, we also included the results of RMSSD and HF. We also added the units on the vertical axes on the graphs in figure 1.
Surely, all patients had depression, therefore we replaced the expression “patients with MDD” with “our patients”. Of course, at the final evaluation, the 19 patients who did not respond to SSRI therapy and were switched to other antidepressants were excluded from the analysis. We specified that now in the text.
We rechecked our data and employed the correlation test again and the reported value of r=0.77 is correct. At this value we have a high (strong) positive correlation between TO and MADRS (only for values greater than 0.9, the correlation is very strong). So this value does not seem too high. Another thing to consider is that, because of a non-Gaussian distribution of our data, we had to employ the Spearman correlation test, which assesses monotonous relationships. In this test the numerical values of MADRS and TO are converted into ranks, then tested. It should not be confused with Pearson’s correlation, which evaluates the linear relationship of the variables.
Best regards,
The authors
Round 2
Reviewer 1 Report
I believe the re-submitted draft of this paper is satisfactory and presents significant research findings very coherently. The topic is of considerable significance and this research offers a valuable contribution.
Author Response
Dear Reviewer,
Thank you very much for your positive review of our article. We are pleased that you are now satisfied with the changes.
Best regards,
The authors
Reviewer 2 Report
Thanks for responding to my previous comments/concerns. Just a couple of final thoughts:
I appreciate the authors mentioning their methods for post-processing the saw ECG signals in the response to review, but it would also be good to include this information in the manuscript. These are important methodological details that sadly often get omitted from psychophysiology papers.
Though the English grammar is largely fine, there are quite a few places where the grammar is a little off. If English is not any of the authors’ first language, I recommend having the manuscript proof-read by a native English speaker.
Author Response
Dear Reviewer,
We followed your suggestion and mentioned in the main text the methods of post-processing the raw ECG data. We also revised the grammar of our article.
Kind regards,
The authors